# Effect of Temperatures Used in Food Storage on Duration of Heat Stress Induced Invasiveness of *L. monocytogenes*

**DOI:** 10.3390/microorganisms7100467

**Published:** 2019-10-17

**Authors:** Ewa Wałecka-Zacharska, Jakub Korkus, Krzysztof Skowron, Magdalena Wietlicka-Piszcz, Katarzyna Kosek-Paszkowska, Jacek Bania

**Affiliations:** 1Department of Food Hygiene and Consumer Health Protection, Wrocław University of Environmental and Life Sciences, 50-375 Wrocław, Poland; jak.kor.95@gmail.com (J.K.); katarzyna.kosek-paszkowska@upwr.edu.pl (K.K.-P.); jacek.bania@upwr.edu.pl (J.B.); 2Department of Microbiology, Nicolaus Copernicus University in Toruń, Collegium Medicum of L. Rydygier in Bydgoszcz, 85-094 Bydgoszcz, Poland; skowron238@wp.pl; 3Department of Theoretical Foundations of Biomedical Sciences and Medical Computer Science, Nicolaus Copernicus University in Toruń, L. Rydygier Collegium Medicum in Bydgoszcz, 9 M. Skłodowska-Curie St., 85-094 Bydgoszcz, Poland; mpiszcz@cm.umk.pl

**Keywords:** *Listeria monocytogenes*, heat stress, invasiveness, food storage temperature

## Abstract

The unpropitious conditions of the food processing environment trigger in *Listeria monocytogenes* stress response mechanisms that may affect the pathogen’s virulence. To date, many studies have revealed that acid, osmotic, heat, cold and oxidative stress modify invasiveness of *L. monocytogenes*. Nonetheless, there is limited data on the duration of the stress effect on bacterial invasiveness. Since most food is stored at low or room temperatures we studied the impact of these temperatures on the duration of heat stress effect on invasiveness of 8 *L. monocytogenes* strains. Bacteria were heat-treated for 20 min at 54 °C and then incubated at 5 and 20 °C up to 14 days. A decrease in invasiveness over time was observed for bacteria not exposed to heating. It was found that heat shock significantly reduced the invasion capacity of all strains and the effect lasted between 7 and 14 days at both 5 and 20 °C. In conclusion, 20-min heating reduces invasion capacity of all *L. monocytogenes* strains; however, the stress effect is temporary and lasts between 7 and 14 days in the food storage conditions. The invasiveness of bacteria changes along with the incubation time and is temperature-dependent.

## 1. Introduction 

*Listeria monocytogenes* is a Gram-positive, non-spore forming, rod-shaped bacterium widespread in the environment [1]. The bacterium is resistant to many environmental factors and is able to grow in a wide range of pH (5.5–9.5), temperatures (0.1–45 °C) and at high salinity (10–20% NaCl) [2]. *L. monocytogenes* is a dangerous food-borne pathogen responsible for listeriosis. The bacterium is an intracellular pathogen able to cross three main host barriers: intestinal, materno–fetal and brain–blood barrier. *L. monocytogenes* may also invade into nonphagocytic cells and spread from cell to cell. It may cause stillbirth, miscarriage, septicemia, meningitis, especially in pregnant women, elderly and immunocompromised people [3]. The incidence of listeriosis varies from 0.1 to 11.3 cases per million people but the mortality rate may range from 10% to 30%. According to European Food Safety Authority (EFSA) this is the highest mortality rate from all food-borne diseases (EFSA, 2018) [4]. In the food processing environment as well as during food preparation and host organism colonization *L. monocytogenes* encounters many adverse factors which elicit in the pathogen stress response mechanisms. One of them is based on an alternative sigma factor B (σ^B^), which allows synthesis of proteins ensuring survival in the deleterious environment. This factor controls expression of around 140 stress-associated genes in *L. monocytogenes* [2]. Moreover, it affects regulation of virulence genes, i.e., *inlAB*, *bsh*, *prfA* [5,6]. Virulence of *L. monocytogenes* strains seems to be heterogenic. It is estimated that up to 21% of *L. monocytogens* population is weakly virulent or avirulent [7]. It has been demonstrated that lineage I strains more effectively invade and spread in epithelial cells and have shorter intracellular generation time than strains of lineage II [8]. There is ample evidence that stress conditions may modify virulence of *L. monocytogenes*. To date, it has been found that low pH of the environment (pH 5–5.5), salt additives and disinfectants induce expression of virulence genes and change the invasion capacity of the pathogen [9,10,11,12]. In addition, from all *L. monocytogenes* serotypes, 4b serotype strains were found to be the most invasive in response to salt stress [9]. Also, temperature of the environment has been shown to have an impact on *L. monocytogenes* invasiveness. It has been demonstrated that low temperatures in most cases increased invasion ability [13,14,15], whereas high temperature decreased the invasiveness [16]. Nonetheless, the effect of stress on bacterial virulence is presumably transient. Our recent study has revealed that the heat-induced invasiveness change lasts on average 32 h in *L. monocytogenes* strains when bacteria are kept at optimal growth temperature (37 °C) after stress exposure [17]. There is practically no data on the duration of stress-induced alterations of *L. monocytogenes* invasiveness at temperatures used in the food storage. The aim of the present study was to assess the influence of food storage temperatures (5 and 20 °C) on the heat-stress effect duration on *L. monocytogenes* invasiveness.

## 2. Materials and Methods 

### 2.1. L. monocytogenes Strains

The study was conducted on 8 *L. monocytogenes* representing 4 serotypes, 3 lineages and 2 sources of isolation (Table 1). Serotyping was performed by the slide agglutination method (Denka Seiken, Japan) and confirmed by multiplex PCR [18]. Lineage assignment was performed according to convention proposed by Zhou et al. (2005) [8] based on C-terminal ActA polymorphism (amino-acids 422–604) inferred from analysis of 599-bp fragment of *actA* gene. The clinical strains derived from patients hospitalized during 2000–2002 in Warsaw and Gdańsk were kindly provided by J. Paciorek (National Institute of Hygiene, Warsaw, Poland).

### 2.2. Growth of L. monocytogenes Strains

Single colonies of bacteria were seeded into 5 mL of BHI broth and were incubated at 37 °C at 230 rpm for 6 h. Then 10 μL of bacterial suspensions were transferred into 4 tubes containing 7.5 mL of fresh BHI and grown for another 18 h. To evaluate the effect of storage temperature on heat-stress induced invasiveness 1.5 mL of bacterial suspensions were transferred into 5 sterile 2 mL Eppendorf tubes and incubated for 20 min at 54 °C in the thermostatic water bath. Additionally, the temperature of water was controlled with a certified thermometer. Then heat-treated bacteria from all 5 tubes, without cooling, were transferred into one tube and together with non-treated bacteria (control) were incubated for 0, 3, 7 and 14 days at 5 or 20 °C. Serial 10-fold dilutions of bacterial suspensions were made and 100 µL of appropriate dilution, resulting in 2 to 5 log CFU (colony forming units) of bacteria, was used to infect HT-29 cell line and was plated onto BHI agar in duplicate.

### 2.3. Cell Line and Culture Conditions

The invasiveness of *L. monocytogenes* was assessed in the human adenocarcinoma cell line HT-29 (CLS, Eppelheim, Germany). Cells were cultured in DMEM (Dulbecco’s modified Eagle’s medium; Sigma-Aldrich, Poznań, Poland) supplemented with 10% heat-inactivated fetal calf serum (FCS) (Invitrogen, Warsaw, Poland), 2 mM glutamine, 100 IU/mL penicillin, and 100 µg/mL streptomycin (Sigma-Aldrich, Poznan, Poland) at 37 °C in 5% CO_2_.

### 2.4. Plaque Forming Assay

All tests were performed on HT-29 cells between passage 10–20. Cells (1 × 10^6^/well) were seeded into 6-well plates 3 days before the test. One day before the test medium was replaced by DMEM supplemented with FCS but without antibiotics. The procedure was repeated 1 h before the test. Next, HT-29 cells were infected with 2 to 5 log CFU of heat-treated and non-treated bacteria, and incubated for 0, 3, 7 and 14 days at 5 or 20 °C. After 2 h wells were washed twice with sterile PBS (Sigma-Aldrich), DMEM supplemented with FCS, containing 100 μg/mL of gentamicin, was added and plates were incubated for another 1.5 h. Then wells were washed again with sterile PBS and finally were overlaid with medium containing FCS, 10 μg/mL of gentamicin and 1.2% of low melt agarose. After 2 days of incubation at 37 °C in 5% CO_2_ the number of plaques was counted manually by visual inspection. Each test was performed at least three times in duplicate. Invasion capacity was calculated as a number of plaques per number of log CFU of viable bacteria at time of infection, deposited per well, and expressed as a percentage.

### 2.5. Statistical Analysis

The summary statistics for continuous variables are presented as mean and standard deviation (SD). Differences between continuous variables were analyzed by the t test for independent samples or by ANOVA together with the Benjamini–Hochberg type adjustment for multiple testing. To study the dependence between the number of bacteria and applied treatment, strain origin, temperature and storage time, the Linear Mixed Effects Model (LMM) has been applied. Initially all the considered factors as well as interactions terms between the considered variables were included in the model as covariates. The backward elimination future selection procedure was applied to find the most significant subset of predictor variables. To analyze the dependence between the bacterial invasiveness and applied treatment, strain origin, temperature and storage time, the Linear Mixed Effects Model (LMM) has also been applied. 

The results were considered as statistically significant when the p-value was less than 0.05. The statistical analysis was performed with the use of the R-software (packages lme4 and gls).

## 3. Results

### 3.1. Assessment of Bacterial Count During Storage at 5 and 20 °C

Bacteria exposed to 20-min heating at 54 °C together with control bacteria, not subjected to heat stress, were incubated at 5 and 20 °C and their number after 0, 3, 7 and 14 days was calculated. The comparison of the number of bacteria for various strains is summarized, both for heat-treated (HT) and non-treated (NT), in Figure 1. Obtained results showed that the number of reisolated bacteria was strain independent both at 5 and 20 °C. The number of bacterial count for strains L4 and L71 was significantly lower than for other strains.

The heat stress treatment (HT) significantly decreased the number of bacteria incubated at 5 and 20 °C (Figure 2). No significant difference in the number of non-treated (NT) bacteria stored at 5 and 20 °C was observed. The mean (SD) of bacterial count HT vs. NT stored at 5 °C was 7.82 (0.46) vs. 9.22 (0.18), *p* < 0.001, and the mean (SD) of bacterial count HT vs. NT stored at 20 °C was 8.19 (0.48) vs. 9.07 (0.48), *p* < 0.001. The statistically lower number of bacteria after heat stress was found at 5 than 20 °C (Figure 2).

Comparing the number of bacterial counts of clinical strains and strains isolated from food, significantly higher values were shown for clinical strains at 20 °C (mean (SD): 8.77 (0.57) vs. 8.55 (0.69), *p* = 0.019). However, no dependence of bacterial count and the strain origin was observed for bacteria stored at 5 °C (Figure 2).

The changes of bacterial counts during storage at 5 and 20 °C are shown in Figure 3. No significant differences in the number of bacteria were found at 5 °C, whereas at 20 °C a significant decrease in the bacteria number was noted after 14 days of storage (mean (SD) = 8.25 (0.69)) as compared to days 0, 3 and 7 (mean (SD) respectively was equal to 8.72 (0.64), 8.79 (0.62), 8.75 (0.53), *p* < 0.001).

On Figure 4 the number of bacteria for consecutive time points for HT and NT bacteria, stored at 5 and 20 °C is presented. For HT bacteria stored at 5 and at 20 °C significant differences in the number of bacteria were seen after 3 and 7 days; the mean (SD) were equal to 7.93 (0.51) vs. 8.35 (0.55), *p* = 0.008, and 7.51 (0.24) vs. 8.4 (0.36), *p* < 0.001, respectively. After 0 and 14 days of incubation no significant differences in the number of bacteria stored at temperature 5 and 20 °C were observed. For bacteria NT stored at 5 and 20 °C significant differences in the number of bacteria were found after 14 days (mean (SD): 9.2 (0.14) vs. 8.68 (0.65), *p* = 0.001). The greatest reduction in bacteria number was obtained after 7 days, for bacteria HT and incubated at 5 °C.

In order to find the independent factors associated with the number of bacteria, the Linear Mixed Effect Model has been applied for data analysis. Table 2 contains the estimates of the final model. The application of heat treatment significantly decreased the number of bacteria. The longer incubation time was also associated with the decrease of bacterial number. The impact of a storage temperature depended on the applied treatment. In non-treated bacteria the temperature of 5 °C was associated with the higher bacterial counts than the temperature of 20 °C. In case of the application of heat treatment and the storage temperature of 5 °C, the number of bacteria was decreased, because of the significant interaction term between treatment and temperature.

### 3.2. Duration of Heat Stress Effect on Invasiveness of L. monocytogenes

To determine how long the effect of heat stress on bacterial invasiveness lasts at temperature used in the food storage, the bacteria were heated for 20 min and then were incubated up to 14 days together with non-treated (control) bacteria at 5 or 20 °C.

The comparison of the invasiveness of both heat-treated and non-treated bacteria for various strains is summarized in Figure 5. Obtained results showed that the invasiveness of *L. monocytognes* was strain independent at temperature of 20 °C, whereas at 5 °C strains L41 and L84 were significantly more invasive than other strains.

Figure 6 presents the invasiveness of HT and NT bacteria, originating from clinical materials and food, incubated at temperature 5 and 20 °C. The invasiveness of the studied bacteria differed statistically significant depending on the storage temperature. Regardless of the treatment, the invasiveness was higher at 5 °C.

Heat stress significantly decreased the invasiveness of bacteria (Figure 6). The invasiveness of HT and NT bacteria stored at 5 °C was equal (mean (SD)) 1.3 (1.22) vs. 2.24 (1.04), *p* < 0.001, respectively, as well as for bacteria stored at 20 °C was (mean (SD)) 0.65 (0.68) vs. 1.22 (1.1), *p* < 0.001, respectively. The invasiveness of HT bacteria stored at 5 °C was on the level of NT bacteria stored at 20 °C.

The invasiveness of clinical strains incubated at 5 °C was significantly higher than the invasiveness of isolates from the food—the mean (SD) respectively was equal to 2.2 (1.49) vs. 1.51 (0.95), *p* = 0.001. No statistical difference between food and clinical strains was found at 20 °C (Figure 6).

On Figure 7 the invasiveness of bacteria during a storage at 5 and 20 °C is summarized. The invasiveness changed along with time. At 5 °C a significantly higher invasiveness was noted after 7 and 14 days compared with 0 and 3 days. At 20 °C a significant decrease of invasiveness after 3, 7 and 14 days compared with time 0 was observed. The lowest invasiveness was registered after 7 days and was statistically significant from the values after 0 and 3 days.

On Figure 8 the evolution of invasiveness for HT and NT bacteria during a storage at 5 °C and 20 °C is depicted. The heat stress significantly reduced the invasiveness of bacteria. The mean(SD) was 2.44 (1.05) vs. 0.45 (0.19), *p* < 0.001. For HT bacteria, the invasiveness initially slightly decreased. This effect was maintained for three days at 5 °C and 7 days at 20 °C. After this time, the invasiveness of the strains began to rise well above the baseline and after 14 days of storage was significantly higher than the invasiveness of NT bacteria the effect of stress on the invasiveness lasted between 7 and 14 days at both 5 and 20 °C. In case of NT bacteria, a downward trend was visible throughout the whole experiment. After 14 days of a storage the invasiveness of NT bacteria was significantly lower for strains stored at 20 °C. For HT bacteria a significant difference in invasiveness between bacteria stored at 5 and 20 °C was found after 7 and 14 days. The mean(SD) was 1.68 (0.6) vs. 0.33 (0.15), *p* < 0.001, and 2.69 (1.35) vs. 1.44 (0.96), *p* = 0.001, respectively. For NT bacteria a significant difference in invasiveness between bacteria stored at 5 and 20 °C was observed after 3 days (mean (SD): 2.2 (0.83) vs. 1.61 (0.83), *p* = 0.018), 7 days (mean (SD): 2.51 (1.23) vs. 0.63 (0.2), *p* < 0.001) and 14 days (mean(SD): 1.82 (0.94) vs. 0.21 (0.15), *p* < 0.001).

In order to select the independent factors associated with bacterial invasiveness the Linear Mixed Effects Model has been fitted to the data. Table 3 contains the estimates of the final model. The application of heat stress significantly decreased the invasiveness of bacteria. The observed effect depended on the storage temperature. The longer storage time of NT bacteria also decreased the invasiveness. In case of the application of HT the longer storage time increased the invasiveness because of the significant interaction term between the stress treatment and the time of storage. The invasiveness of both NT and HT bacteria was significantly higher at 5 °C as compared to 20 °C.

## 4. Discussion

During food production and storage *L. monocytogenes* is constantly exposed to deleterious environmental conditions that affect its growth. To survive in such environment, bacteria change their metabolism, inducing stress response mechanisms. The stress response may result in increased resistance to another adverse factor but also influence bacterial virulence. So far, it has been shown that osmotic, acid, cold and heat stress, as well as high pressure may modify the invasiveness of *L. monocytogenes* [19]. Finally, the stress-induced changes are eliminated and the homeostasis is restored [20,21]. Nevertheless, there is limited data on how long the stress-induced alterations, including changes of the invasiveness, can be observed in the food storage conditions.

In this work the impact of heat stress on invasiveness of eight *L. monocytogenes* strains and its duration at food storage temperatures was assessed. To elicit the stress response the bacteria were exposed to heating at 54 °C. This is the average temperature which the pathogen may experience during reheating of food products or in undercooked food [22]. This temperature allows the reduction of bacteria number, but the number of survived bacteria is still high enough to determine changes of the invasiveness in response to stress [17].

Cooling is a very important stage for the food safety assurance. The food of high risk, perishable food as well as cooked food, but not consumed immediately, should be cooled as quickly as possible and stored in the refrigerator at 5 °C or less [23]. Not all food products have to be kept refrigerated to be safe. Bread, dry products or canned food should be stored at room temperature (19–20 °C) [24]. Therefore, in the present study the duration of heat-stress induced invasiveness of *L. monocytogenes* strains was determined at 5 and 20 °C. The human adenocarcinoma HT-29 cell line with plaque forming assay were used to assess invasion capacity of the bacteria. Over the years, many methods have been developed to study *L. monocytogenes* virulence [25]. These include in vivo tests using laboratory animals (mice, guinea pigs, monkeys) or invertebrates (*Caenorhabditis elegans*, *Drosophila melanogaster*, Zebra fish, *Galleria mellonella*) and in vitro cell assays (enterocyte-like cell line Caco-2, human adenocarcinoma HT-29, hepatocyte Hep-G2, macrophage-like J774) [25,26]. Mammalian models are regarded as a gold standard for the assessment of pathogen virulence but are associated with ethical burdens and high costs. Invertebrates are easy to manipulate and inexpensive but not all can survive at human body temperature and may involve difficulties in precise delivery of the inoculum. The main benefits of in vitro cell assays are their simplicity, low cost, rapid growth and availability of human cells. However, this type of model does not necessarily reflect physiology in vivo and time of infection is limited by bacterial overgrowth. In such tests the pathogen colonization is restricted to only one type of cells and the ability to evaluate host response is limited [27]. Cell assays exclude also passage through gastrointestinal tract that is a great challenge for the pathogen. However, plaque forming assay has been demonstrated to discriminate between virulent, hypovirulent and avirulent strains of *L. monocytogenes* and is considered as the best alternative for in vivo tests to study *L. monocytogenes* virulence. This assay, in contrast to commonly used invasion assay, determines not only the ability to invade the human cells but also the ability to spread to adjacent cells [28]. Since invasion of epithelia by *L. monocytogenes* has been found to be affected by the cell proliferation rate in this study we used HT-29 cell line that is characterized by a constant proliferation rate [29].

In this study eight strains of *L. monocytogenes*, exposed to 20-min heating at 54 °C, were incubated together with non-treated bacteria at 5 and 20 °C for 0, 3, 7 and 14 days and subsequently the invasion capacity was determined. The heat exposure significantly affected the survival and the invasiveness of bacteria. This supports our earlier study [17]. In the present study, the invasiveness of bacteria changed along with the incubation time and was temperature-dependent. The invasiveness of heat-treated bacteria initially slightly decreased and then, after 3 days at 5 °C and 7 days at 20 °C, started to rise. The effect of heat stress on *L. monocytogenes* invasiveness lasted between 7 and 14 days both at 5 and 20 °C. After two weeks of the storage at both temperatures the invasiveness of heat-treated bacteria was significantly higher than the invasion capacity of non-treated bacteria. Our previous study assessing the duration of heat-induced *L. monocytogenes* invasiveness revealed that in 80% of strains it lasted on average 32 h at 37 °C [15]. Hence, it can be concluded that the duration of heat stress effect extends together with a decrease of temperature to the room temperature. Further decrease in temperature to cold storage temperature does not affect the duration of the observed effect. In contrast, Stollework et al. (2017) [30] demonstrated that a 10-min heating at 55 °C did not change invasiveness of bacteria during a 14-day storage at 8 °C. On the other hand, bacteria subjected to high pressure processing decreased the invasiveness but returned to the rate of non-treated bacteria after 14 days of storage at 8 °C.

In this work the invasiveness of bacteria at 20 °C was strain-independent and no correlation between strain’s origin and heat-stress induced invasiveness was found. On the other hand, at 5 °C, two strains were significantly more invasive than the rest of the strains as well as heat-treated clinical strains compared with the treated strains of food origin.

The invasion ability of non-treated strains significantly decreased during a storage at 20 °C, whereas at 5 °C they remained at the similar level. Our earlier study showed that in 60% of non-treated strains invasiveness was reduced along with the incubation at 37 °C [17]. The observed reduction of invasiveness in the non-treated strains might be due to gradual nutrients and oxygen depletion which is a stress factor for bacteria (Rees et al., 1995) [31]. Since in the heat-treated bacteria a stress response was already induced they responded to another adverse factor with increased invasiveness.

Comparing the invasiveness of bacteria stored at 5 and 20 °C, statistically higher values were found at 5 °C. Low temperature is another stress factor for bacteria which may increase their invasion ability [13,14,15]. On the other hand, in such conditions the metabolic rate is decreased and nutrients are consumed slower. This may explain the differences between storage at 5 and 20 °C.

The mechanism underlying the observed alterations of invasiveness during storage is not clearly explicated. It is widely recognized that internalins are key factors allowing invasion into epithelial cells. Expression of *inlAB* genes is controlled by an alternative sigma B factor and the key regulator of virulence genes PrfA [32]. Hence, it can be assumed that the changes of invasiveness could be explained by the level of *inlA*, *inlB*, *prfA* and *sigB* transcripts. Previous studies, however, have reported that expression of internalins does not always correlate with the invasiveness of *L. monocytogenes* [33,34]. Our latest study revealed that increase of the invasiveness was related to upregulation of *inlAB* genes, whereas the decrease of invasiveness was not associated with the level of these transcripts. The link between *sigB* and *prfA* genes, and the invasion capacity was also not found [17]. Therefore, it can be hypothesized that additional factors are involved in this process and further studies are needed to elucidate the observed phenomenon.

## 5. Conclusions

We revealed a timescale of stress-induced alteration in *L. monocytogenes* invasiveness at temperatures encountered by the pathogen during food storage. Our results prove that the invasion capacity of *L. monocytogenes* strains decreases in response to 20-min heat exposure. Both heat-stressed and non-stressed bacteria incubated at 5 °C are more invasive than at 20 °C. The effect of heat-stress on *L. monocytogenes* invasiveness is transient both at 5 and 20 °C. Decline of the stress effect is observed between 7 and 14 days. From the food safety point of view heat treatment reduces the bacterial number; however, the survivors become more invasive.

## Figures and Tables

**Figure 1 microorganisms-07-00467-f001:**
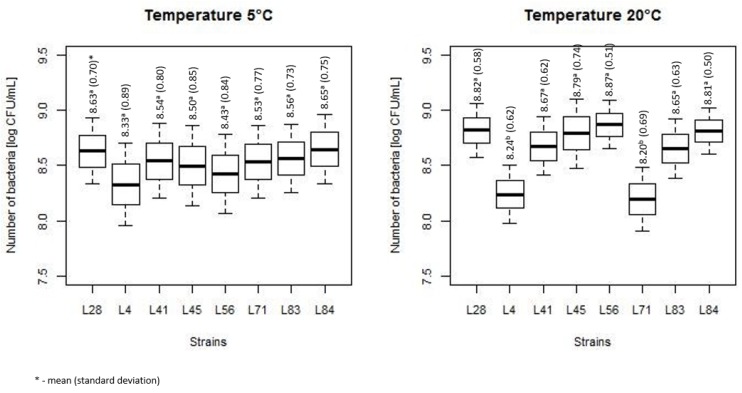
The summarized number of bacteria for various strains (the horizontal line is the mean, the box spans represent standard error and the whiskers represent 95% confidence interval).

**Figure 2 microorganisms-07-00467-f002:**
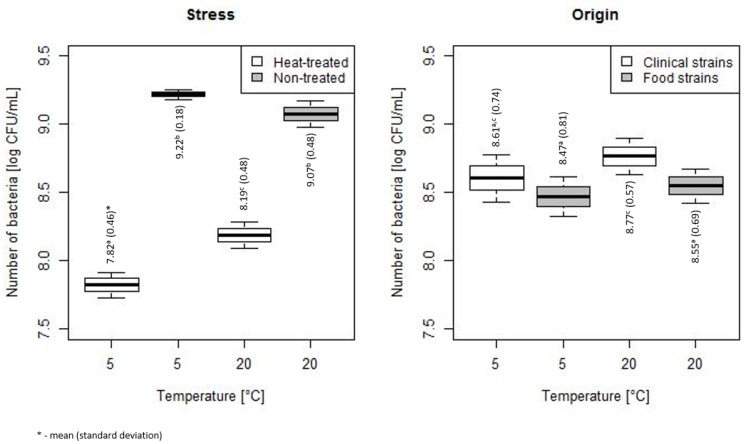
Comparison of bacterial counts of heat-treated (HT) and non-treated (NT) bacteria, and for clinical and food strains (the horizontal line is the mean, the box spans represent standard error and the whiskers represent 95% confidence interval).

**Figure 3 microorganisms-07-00467-f003:**
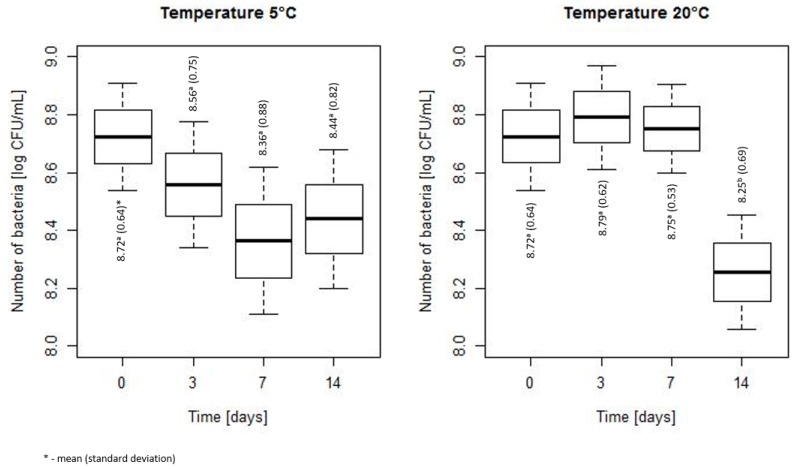
The changes of the bacteria number during storage at 5 and 20 °C (the horizontal line is the mean, the box spans represent standard error and the whiskers represent 95% confidence interval).

**Figure 4 microorganisms-07-00467-f004:**
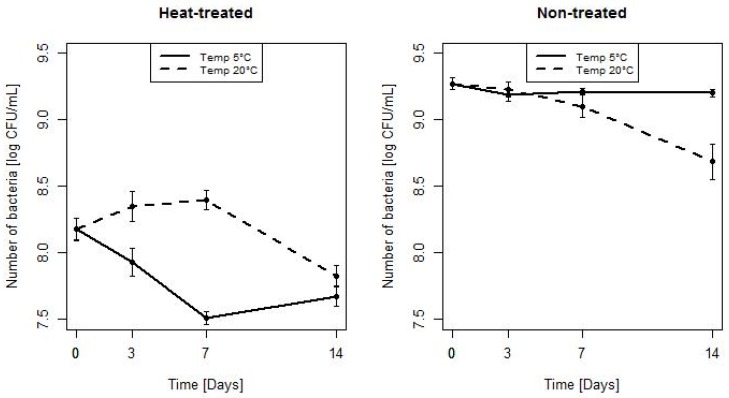
The changes of the mean number of HT and NT bacteria stored at 5 and 20 °C (error bars represent standard errors of the mean).

**Figure 5 microorganisms-07-00467-f005:**
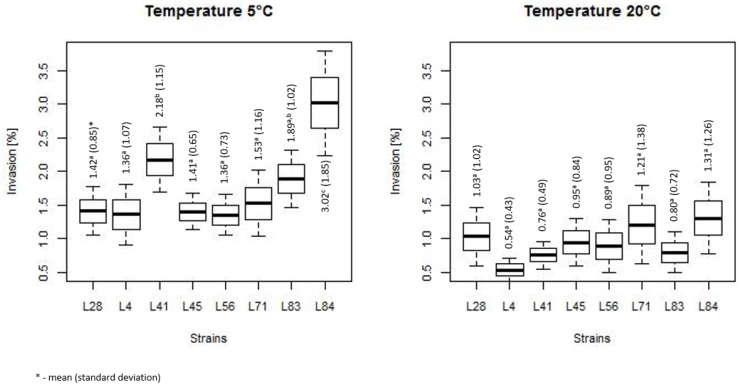
The summarized invasiveness for various strains (the horizontal line is the mean, the box spans represent standard error and the whiskers represent 95% confidence interval).

**Figure 6 microorganisms-07-00467-f006:**
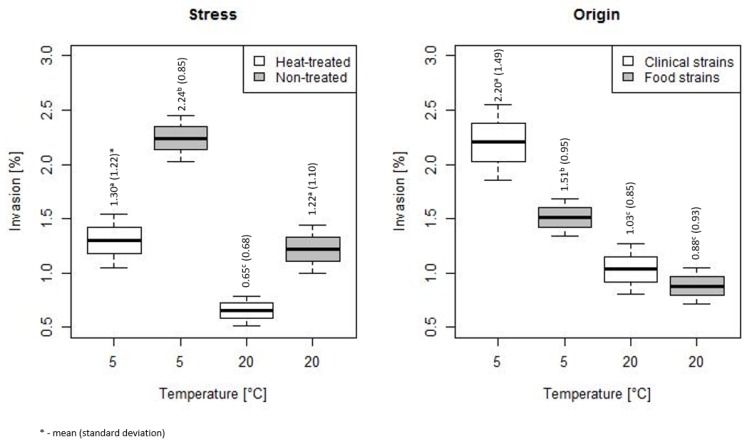
The invasiveness of HT and NT bacteria, and for bacteria from food and clinical samples, stored at 5 and 20 °C (the horizontal line is the mean, the box spans represent standard error and the whiskers represent 95% confidence interval).

**Figure 7 microorganisms-07-00467-f007:**
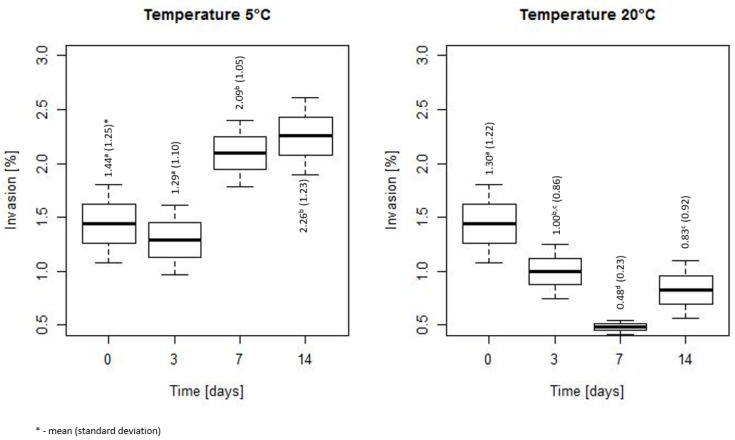
The invasiveness of bacteria during a storage at 5 and 20 °C (the horizontal line is the mean, the box spans represent standard error and the whiskers represent 95% confidence interval).

**Figure 8 microorganisms-07-00467-f008:**
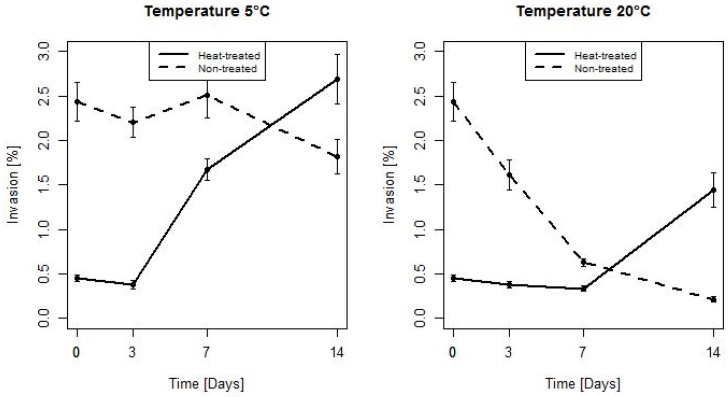
The changes of the invasiveness of HT and NT bacteria stored at 5 and 20 °C (error bars represent standard errors of the mean).

**Table 1 microorganisms-07-00467-t001:** *L. monocytogenes* strains used in the study.

Strain	Source	Lineage	Serotype
L28	Human	I	4b
L84	Human	I	4b
L41	Human	I	1/2b
L45	Food	I	1/2b
L56	Food	II	1/2a
L71	Food	II	1/2a
L83	Food	II	1/2a
L4	Food	III	4c

**Table 2 microorganisms-07-00467-t002:** Predictive factors for the number of bacteria identified by the Linear Mixed Effects Model.

Parameters	Value	Std. Error	*t*-Value	*p*-Value
(Intercept)	9.22	0.07	133.98	<0.001
Stress HT vs. NT	−0.88	0.05	−16.75	<0.001
Temperature 5 °C vs. 20 °C	0.15	0.05	2.78	0.006
Storage (Days) 3 vs. 0	−0.05	0.05	−0.93	0.353
Storage (Days) 7 vs. 0	−0.17	0.06	−2.83	0.005
Storage (Days) 14 vs. 0	−0.38	0.07	−5.1	<0.001
Stress HT: Temperature 5 °C	−0.51	0.07	−6.88	<0.001

**Table 3 microorganisms-07-00467-t003:** Predictive factors for the invasiveness of bacteria identified by the Linear Mixed Effects Model.

Parameters	Value	Std. Error	*t*-Value	*p*-Value
(Intercept)	1.93	0.19	10.2	<0.001
Stress HT vs. NT	−1.8	0.16	−11.06	<0.001
Temperature 5 vs. 20	1.02	0.1	9.9	<0.001
Storage (Days) 3 vs. 0	−0.53	0.16	−3.33	0.001
Storage (Days) 7 vs. 0	−0.87	0.2	−4.24	<0.001
Storage (Days) 14 vs. 0	−1.42	0.32	−4.39	<0.001
Stress HT: Temperature 5	−0.37	0.15	−2.55	0.011
Stress HT: Storage (Days) 3	0.46	0.21	2.22	0.027
Stress HT: Storage (Days) 7	1.43	0.21	6.92	<0.001
Stress HT: Storage (Days) 14	3.04	0.21	14.74	<0.001

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
