# Peer review of "Effect of Temperatures Used in Food Storage on Duration of Heat Stress Induced Invasiveness of L. monocytogenes"

_microorganisms, 2019, doi:10.3390/microorganisms7100467_

Round 1

Reviewer 1 Report

This study builds off previous work from the authors investigating how prolonged is an effect on Lm invasiveness after heat treatment. The study compared storage post-treatment at 5 and 20C vs previous work in Lm cultured at 37C. 

After investigating 8 strains, which showed a high amount of variability, one consistent observation was that heat treatment consistently reduced Lm invasiveness initially, but over time most heat-treated strains became more invasive while non-treated controls became less invasive. Often this resulted in very little biological difference in invasiveness between treated and non-treated strains by 14 days post-heat treatment. Also interestingly, the percent invasion after 14 days in heat treated strains was usually more pronounced in cultures stored at 5C rather than 20C. 

I don't have significant concerns about the controls and experimental design. I did have some notes on clarity, presentation, and phrasing, which I will list by manuscript line.

72-73  Overall I think this sentence would be more helpful in the description of the Plaque-forming assays.  What was the volume via which the 2-5 log Lm were used to infect the HT-29?

81 For clarity, wither "between 10 and 20 passages" or "between passage 10-20"

81-82  I suggest "Cells (1x106) were seed into 6 well plates....

84  Here is where I would like to see or be reminded of infection cfu

90  So for each infection, the number of plaques formed was compared to the CFU of Lm viable at time of infection based on the BHI plates?  And so in the figures Invasion [%] referrs to plaques/BHI cfu?  Given the labels used in the figures, I would make it very clear in this line that we're looking at percents, since there was variable viability of the Lm strains over time in the different storage conditions.

98  You hit a numbering inconsistency at this point. This should be section 3.1, rather than 2.1, since the methods was section 2.  See also lines 120, 155, 179, 194, 

100  number at Day 0, 3, 7, and 14.  Or, "after 0, 3, 7, and 14 days"

110  You mention significant differences in the number of bacteria.  Do you mean statistically significant?  If so, Please note this visually in the graphs and mention in the figure legends so that the figure can stand on its own.  If the differences are not statistically significant, please do not use "significant" 

122  awkward. Instead: To determine how long the effect of heat stress on bacterial invasivness lasts 

126 and throughout.  I notice you frequently use numerals  rather than spelling out the numbers. Check with the editor on this, as typically numbers are spelled out if they can be expressed with fewer than 3 digits.  Just a stylistic note.

130-133, as with 110, please incorporate a visual cue or note in the figure legend regard which strains showed statistically significant differences.

148 In this case, you show a visual demarcation of statistical significance, but you don't mention the significance in the text.

226: effect was visible longer at 5C than 20C

Author Response

Poland, August 30, 2019

Response letter to Reviewers

I would like to thank all reviewers for their valuable comments. All modification and new information have been marked in the text with yellow color. According to the suggestions of the reviewer statistical analysis has been changed and ANOVA together with the Benjamini-Hochberg type adjustment for multiple testing and Linear Mixed Effects Model have been applied. Thereby results section has been modified.

Reviewer 1

 Thank you for all valuable comments. All suggestions have been applied in the revised version of the manuscript.

72-73  Overall I think this sentence would be more helpful in the description of the Plaque-forming assays.  What was the volume via which the 2-5 log Lm were used to infect the HT-29?

-  The  volume used was 100 µL. this information was include in the text: line 87.

81 For clarity, wither "between 10 and 20 passages" or "between passage 10-20"

-  Between passage 10-20 : it was modified in the text (line 96)

81-82  I suggest "Cells (1x106) were seed into 6 well plates....

-  The suggestion was included (line 96)

84  Here is where I would like to see or be reminded of infection cfu

- cfu was remainded ( line 99)

90  So for each infection, the number of plaques formed was compared to the CFU of Lm viable at time of infection based on the BHI plates?  And so in the figures Invasion [%] referrs to plaques/BHI cfu?  Given the labels used in the figures, I would make it very clear in this line that we're looking at percents, since there was variable viability of the Lm strains over time in the different storage conditions.

-  Yes plaques formed by L. monocytogenes were compared to CFU of Lm viable at time of infection based on the BHI plates. The invasiveness was calculated as a s a number of plaques per number of log CFU of viable bacteria at time of infection, deposited per well, and expressed as a percentage. This information was added in the text ( lines 105- 107).

98  You hit a numbering inconsistency at this point. This should be section 3.1, rather than 2.1, since the methods was section 2.  See also lines 120, 155, 179, 194, 

- The numbering was corrected.

100  number at Day 0, 3, 7, and 14.  Or, "after 0, 3, 7, and 14 days"

- It was corrected ( line 124)

110  You mention significant differences in the number of bacteria.  Do you mean statistically significant?  If so, Please note this visually in the graphs and mention in the figure legends so that the figure can stand on its own.  If the differences are not statistically significant, please do not use "significant" 

- The statistical analysis has been changed and new figures are included.

122  awkward. Instead: To determine how long the effect of heat stress on bacterial invasiveness lasts 

- It was modified according to the reviewer suggestion ( line 186).

126 and throughout.  I notice you frequently use numerals  rather than spelling out the numbers. Check with the editor on this, as typically numbers are spelled out if they can be expressed with fewer than 3 digits.  Just a stylistic note.

- Now numerals were used only in the expression "out of" e.g. 2 out of 6.

Best regards,

Ewa Wałecka-Zacharska

Reviewer 2 Report

The manuscript explores effect of recovery of heat stressed Listeria monocytogenes strains comparing the invasiveness ability in cell culture. the most important aspect is that the conclusions are well known since many time ago. The authors must clarify the real importance of this research.

Author Response

Poland, August 30, 2019

Response letter to Reviewers

I would like to thank all reviewers for their valuable comments. All modification and new information have been marked in the text with yellow color. According to the suggestions of the reviewer statistical analysis has been changed and ANOVA together with the Benjamini-Hochberg type adjustment for multiple testing and Linear Mixed Effects Model have been applied. Thereby results section has been modified.

Reviewer 2

The effect of stress on invasiveness of L. monocytogenes is widely recognized. However, little was known about duration of stress-induced alterations of invasiveness. Our study revealed that this effect is transient and lasts between 7 and 14 days at the food storage temperature. However, after 2 weeks of a storage bacteria exposed to heat treatment become more invasive than non-treated bacteria.

Best regards,

Ewa Wałecka-Zacharska

Reviewer 3 Report

Walecka-Zacharska et al present a very focused work demonstrating that heat-stress of L. monocytogenes  followed by incubation up to 14 days at refrigeration temperatures has a demonstrable and consistent effect on invasiveness.  In addition, they show that storage at 5C (refrigeration temp) results in increased invasiveness compared with storage at 20C (room temp). The major effects seem to be consistent across the 8 strains tested and it appears that clinical strains have increased invasiveness over food strains when they have been subjected to heating followed by 14 days incubation at 5C. These results, although highly relevant to food safety, could be improved upon substantially by additional work to identify a mechanism for the findings, e.g. via increased expression of prfA, InlA, or σB that could explain the changes in invasion. The paper could also be improved by minor editing and by formatting changes to the figures to make them more reader-friendly.

Specific comments.

Materials and Methods

1) Is there as reference for the Listeria strains used and how the serotypes and lineages were determined?

2) The use of the word “combined” in the sentence beginning Line 70 is confusing. Combined with what?

3) In the description of the plaque-forming assay, please be clear to state if there is/is not fetal calf serum in the DMEM on lines 83, 86, and 87.

4) Line 81 should read “Cells (1,000,000/well) were seeded into 6-well plates…”

5) Line 89, were the numbers of plaques “estimated” or counted? If estimated, the method & its accuracy need to be stated.

6) Line 133, “l83” should be “L83”

Results

1) In Figures and 4 it would be helpful to include symbols on the figure to indicate statistical significance.

2) In Figure 6, additional labeling on the y axis to indicate “non-heated” and “heat-treated” groups, in addition to the color-coding, would make the figure easier to understand.

3) Line 160, “reported” should be “observed”

4) Line 164, “mounted” is not the right word, perhaps “surpassed” is better depending on what the authors intend.

5) Line 183, “between 5C and 20C” is perhaps better written as “at 5C compared with 20C”.

Discussion

1) This is reasonably well-done but without additional data on a mechanism for the findings it is limited in its scope.

Author Response

Poland, August 30, 2019

Response letter to Reviewers

I would like to thank all reviewers for their valuable comments. All modification and new information have been marked in the text with yellow color. According to the suggestions of the reviewer statistical analysis has been changed and ANOVA together with the Benjamini-Hochberg type adjustment for multiple testing and Linear Mixed Effects Model have been applied. Thereby results section has been modified.

Reviewer 3

Thank you for all valuable comments. As far as a mechanism for our findings is consider

we didn't study expression of inlAB, prfA and sigB genes because our previous study has showed that increased inlAB transcripts level correlated with an increase of invasiveness but decrease of invasiveness was not associated with downregulation of genes expression. We have also not observed the link between invasiveness and prfA/sigB transcripts. Also previous studies have found that invasiveness of L. monocytogenes is not always related to  expression of internalins, suggesting the involvement of additional factors. We put such comment in the discussion section (lines 325-335).

The another reason for not investigating the gene expression this time was the duration of the experiment. Since bacteria were stored up to two 2 weeks at 5°C or 20°C the amount of dead cells and the quality of RNA could affect the reliability of the results.

The other reviewer's suggestions  have been applied.

Materials and Methods

1) Is there as reference for the Listeria strains used and how the serotypes and lineages were determined?

No reference strain was used as we used environmental strains, isolated from food and humans, that have been used by us in earlier studies and are well known.

Serotyping was performed by the slide agglutination method and confirmed by multiplex PCR [ Doumith et al. 2004]. Lineage assignment was performed according to convention proposed by Zhou et al. (2005) based on C-terminal ActA polymorphism ( amino-acids 422–604) inferred from analysis of 599-bp fragment of actA gene. This information was included in the text ( lines 71-74).

2) The use of the word “combined” in the sentence beginning Line 70 is confusing. Combined with what?

Indeed, this the sentence might be confusing and was changed into: Then heat-treated bacteria from all 5 tubes, without cooling, were transferred into one tube and together with non-treated bacteria (control) were incubated for 0, 3, 7 and 14 days at 5°C or 20°C (lines 83-85).

3) In the description of the plaque-forming assay, please be clear to state if there is/is not fetal calf serum in the DMEM on lines 83, 86, and 87.

FCS was always added to the medium and this information was included (lines 98, 101, 103)

4) Line 81 should read “Cells (1,000,000/well) were seeded into 6-well plates…”

It was modified into Cells (106/ well) (line 96)

5) Line 89, were the numbers of plaques “estimated” or counted? If estimated, the method & its accuracy need to be stated.

The number of plaques was counted manually by visual inspection. This information was included in the text (line 105 ).

6) Line 133, “l83” should be “L83”

It was corrected.

Results

This section has been modified significantly and new figures were included as different statistical analysis was applied ( according to suggestions of one of the reviewers).

Best regards,

Ewa Wałecka-Zacharska

Reviewer 4 Report

The topic of the paper is interesting and worthy of investigation, but there are some issues to be addressed before publication:

a) materials and methods: there are many details missing (how the temperature was measured during the heat treatment; the duration of the holding time; if the samples were cooled; how PFU were counted etc...)

b) Statistic: the most correct approch is the use of Multivariate Anova to assess the effect of the temperature, duration of the storage and strain. If the authors would like the difference between clinical and not-clinical strains, another categorical predictor should be added. This kind of approach has two main outputs: a table of standardized effect, to point out the post significant variable, and the decomposition of the statistical hypothesis, to see the quantitative effect of each factor. I think that the merit of the paper could be signifincatly increased with an adequate statistical treatment of the results.

c) add in the introduction some details on the invasiveness of L. monocytogenes

d) compare the results with similar published data and try to elucidate the benefits and the limits of an in vitro approach.

Author Response

Poland, August 30, 2019

Response letter to Reviewers

I would like to thank all reviewers for their valuable comments. All modification and new information have been marked in the text with yellow color. According to the suggestions of the reviewer statistical analysis has been changed and ANOVA together with the Benjamini-Hochberg type adjustment for multiple testing and Linear Mixed Effects Model have been applied. Thereby results section has been modified.

Reviewer 4

Thank you for all comments and suggestion.

a) materials and methods: there are many details missing (how the temperature was measured during the heat treatment; the duration of the holding time; if the samples were cooled; how PFU were counted etc...)

-All missing details added in the text (lines 83-85 ).The thermostatic water bath was used. Additionally, the temperature of water was controlled with a certified thermometer. Samples were not cooled but immediately were used f for the experiment (time 0) all put at 5°C or 20°C. The number of plaques was counted manually by visual inspection (line 105).

b) Statistic: the most correct approch is the use of Multivariate Anova to assess the effect of the temperature, duration of the storage and strain. If the authors would like the difference between clinical and not-clinical strains, another categorical predictor should be added. This kind of approach has two main outputs: a table of standardized effect, to point out the post significant variable, and the decomposition of the statistical hypothesis, to see the quantitative effect of each factor. I think that the merit of the paper could be signifincatly increased with an adequate statistical treatment of the results.

- According to the suggestions statistical analysis has been changed and ANOVA together with the Benjamini-Hochberg type adjustment for multiple testing and Linear Mixed Effects Model have been applied. Thereby results section has been modified.

c) add in the introduction some details on the invasiveness of L. monocytogenes

Details on L. monocytogenes invasiveness were added:

The bacterium is an intracellular pathogen able to cross three main host barriers: intestinal, materno-fetal and brain-blood barrier. L. monocytogenes may also invade into nonphagocytic cells and spread from cell to cell. (lines 39-42 )

Virulence of L. monocytogenes strains seems to be heterogenic. It is estimated that up to 21% of L. monocytogens population is weakly virulent or avirulent [Roche 2005 ]. It has been demonstrated that lineage I strains more effectively invade and spread in epithelial cells, and have shorter intracellular generation time than strains of lineage II[ Zhou 2005 ].(lines )

In addition, from all L. monocytogenes serotypes 4b serotype strains were found to be the most invasive in response to salt stress [9]. (lines 51-54 )

d) compare the results with similar published data and try to elucidate the benefits and the limits of an in vitro approach.

The benefits and limits of use in vitro tests compared to in vivo tests were added in the discussion section:

Over the years, many methods have been developed to study L. monocytogenes virulence [Liu 2006]. This include in vivo tests using laboratory animals (mice, guinea pigs, monkeys) or invertebrates (Caenorhabditis elegans, Drosophila melanogaster, Zebra fish, Galleria mellonella) and in vitro cell assays (enterocyte-like cell line Caco-2, human adenocarcinoma HT-29, hepatocyte Hep-G2, macrophage-like J774). Mamalian models are regarded as a gold standard for the assessment of pathogen virulence but are associated with ethical burdens and high costs. Invertebrates are easy to manipulate and inexpensive but not all can survive at human body temperature and may involve difficulties in precise delivery of the inoculum. The main benefits of in vitro cell assays is their simplicity, low cost, rapid growth and availability of human cells. However, this type of model  not necessarily reflects physiology in vivo and time of infection is limited by bacterial overgrowth. In such tests the pathogen colonization is restricted only to one type of cells and the ability to evaluate host response  is limited. Cell assays exclude also passage through gastrointestinal tract that is a great challenge for the pathogen.  However, plaque forming assay has been demonstrated to discriminate between virulent, hypovirulent and avirulent strains of L. monocytogenes and is considered as the  best alternative for in vivo tests to study L. monocytogenes virulence. This assay, in contrast to commonly used invasion assay, determines not only the ability to invade into the human cells but also the ability to spread to adjacent cells (lines 275-291).

Best regards,

Ewa-Wałecka-Zacharska

Round 2

Reviewer 2 Report

Some aspects of the manuscript have been improved, however the main flaws of the manuscript are the same. So my recommendation continues to be of rejecting the manuscript, because the conclusions are well known, and it is needed a major advance in knowledge for a journal of impact 4.

Reviewer 4 Report

the authors addressed my issues; therefore I recommend the acceptance of the manuscript